

# Identification of species and materia medica within *Saussurea* subg. *Amphilaena* based on DNA barcodes

Jie Chen, Yong-Bao Zhao, Yu-Jin Wang and Xiao-Gang Li

State Key Laboratory of Grassland Agro-Ecosystem, School of Life Sciences, Lanzhou University, Lanzhou, Gansu, China

## ABSTRACT

*Saussurea* is one of the most species-rich genera in the family Asteraceae, where some have a complex evolutionary history, including radiation and convergent evolution, and the identification of these species is notoriously difficult. This genus contains many plants with medical uses, and thus an objective identification method is urgently needed. *Saussurea* subg. *Amphilaena* is one of the four subgenera of *Saussurea* and it is particularly rich in medical resources, where 15/39 species are used in medicine. To test the application of DNA barcodes in this subgenus, five candidates were sequenced and analyzed using 131 individuals representing 15 medical plants and four additional species from this subgenus. Our results suggested that internal transcribed spacer (ITS) + *rbc*L or ITS + *rbc*L + *psb*A-*trn*H could distinguish all of the species, while the ITS alone could identify all of the 15 medical plants. However, the species identification rates based on plastid barcodes were low, i.e., 0% to 36% when analyzed individually, and 63% when all four loci were combined. Thus, we recommend using ITS + *rbc*L as the DNA barcode for *S.* subg. *Amphilaena* or the ITS alone for medical plants. Possible taxonomic problems and substitutes for medicinal plant materials are also discussed.

## INTRODUCTION

*Saussurea* is one of the most species-rich genera in Asteraceae and the taxonomic identification of these species is notoriously difficult (*Lipschitz, 1979*). Recent radiation, widespread hybridization, and convergent evolution have combined to make the delimitation of these species extremely complicated (*Wang et al., 2009*). Among the 289 recognized species in the "Flora of China" (FOC), many are very challenging to differentiate, with one or several morphologically similar species (*Shi & Raab-Straube, 2011*). For example, about nine current widely accepted species are suspected to be conspecific with *S. taraxacifolia* (*Chen, 2015*). Since the publication of FOC, the newly described species have totaled more than 60 species (*Chen, 2015*; *Wang et al., 2014*; *Xu, Hao & Xia, 2014*; *Chen & Wang, 2018*), with an average of 10 species every year, which is a far higher number than that of other genera. These new species have mostly been separated from the known species and at least 10 of them bear the prefix "pseudo" to indicate their similarity in terms of morphology (*Chen, 2014*; *Chen & Yuan, 2015*; *Wang et al., 2014*).

Corresponding author
Yu-Jin Wang, wangyujin@lzu.edu.cn

This taxonomic problem particularly affects *S.* subg. *Amphilaena*, which is one of the four subgenera of *Saussurea*, where these species are defined mainly based on the self-transparent and colorful bract that subtends the synflorescence (Fig. 1) (*Lipschitz, 1979*; *Raab-Straube, 2017*). This character is a well-known adaptation to high altitudes and it occurs in a number of angiosperm genera from different families (*Omori, Takayama & Fls, 2000*). Within *S.* subg. *Amphilaena*, it has also been documented that this character was derived multiple times and some of the species showing very high similarity, such as *S. involucrata* and *S. obvolata*, are actually distantly related according to molecular phylogeny (*Wang et al., 2009*). In addition, this subgenus is considered to be a result of a recent radiation in the Qinghai–Tibet Plateau where 35 of the total number of 38 species have been recorded (*Raab-Straube, 2017*). This type of process usually produces many closely related species where one species might resemble several other species, thereby yielding a number of complexes (*Simões et al., 2016*).

Complex taxonomy undoubtedly causes problems with identification, and among the 38 species recognized in the latest monograph, at least 13 species are widely misidentified. For example, *S. orgaadayi* was long misidentified as *S. involucrata* (*Smirnov, 2004*), although both species were described many years ago and the latter is one of the most famous plants in China because of its beauty and usage in traditional Chinese medicine (*Chik et al., 2015*). In addition, eight species within the *S. obvallata* complex have been recognized as single species since the establishment of *S. obvallata* (*Raab-Straube, 2017*).

Evidently, misidentification can lead to a misunderstanding of biodiversity. In some cases, these errors can even be deadly harmful for humans given that many *Saussurea* species are used in medicine (*Chik et al., 2015*; *Li, Zhu & Cai, 2000*; *Yang et al., 2005*). In addition to *S. involucrata*, 14 other species have been formally recorded as medically useful in *S.* subg. *Amphilaena* (Table 1) (*Cao et al., 2016*; *Chen, Pei & Zhao, 2010*; *Jiang, Luo & Xu, 2010*; *Li, 1999*). However, the authentication of species is time-consuming and it requires a specialist taxonomist in most cases. Moreover, some species are found only in areas that are difficult to access, possibly because of their excessive consumption. For example, *S. involucrata* is currently listed as second-class protected plants due to over-exploitation (*Fu & Jin, 1992*), while *S. wettsteiniana* and *S. velutina* are both endemic to a few mountains in Sichuan, China, and they are difficult to obtain due to their restricted distributions (*Shi & Raab-Straube, 2011*). Thus, possible substitutes for these species are urgently needed to be ascertained.

DNA barcoding is a rapid and reliable technique for identifying species based on variations in the sequence of short standard DNA regions. Phylogenetic studies based on these fragments can also help to identify substitute plants. However, the selection of the fragments used for DNA barcoding is a controversial problem. The Plant Working Group of the Consortium for the Barcode of Life (CBOL) proposed using a combination of *rbc*L and *mat*K as a "core barcode" for identifying land plants (*Hollingsworth et al., 2009*). Subsequently, *trn*H-*psb*A and the nuclear ribosomal internal transcribed spacer (ITS) were proposed as supplementary barcodes for land plants (*Kress et al., 2005*; *Li et al., 2011*). In addition, *trn*K was found to outperform *mat*K in some studies (*Cao et al., 2010*; *Müller & Borsch, 2005*).

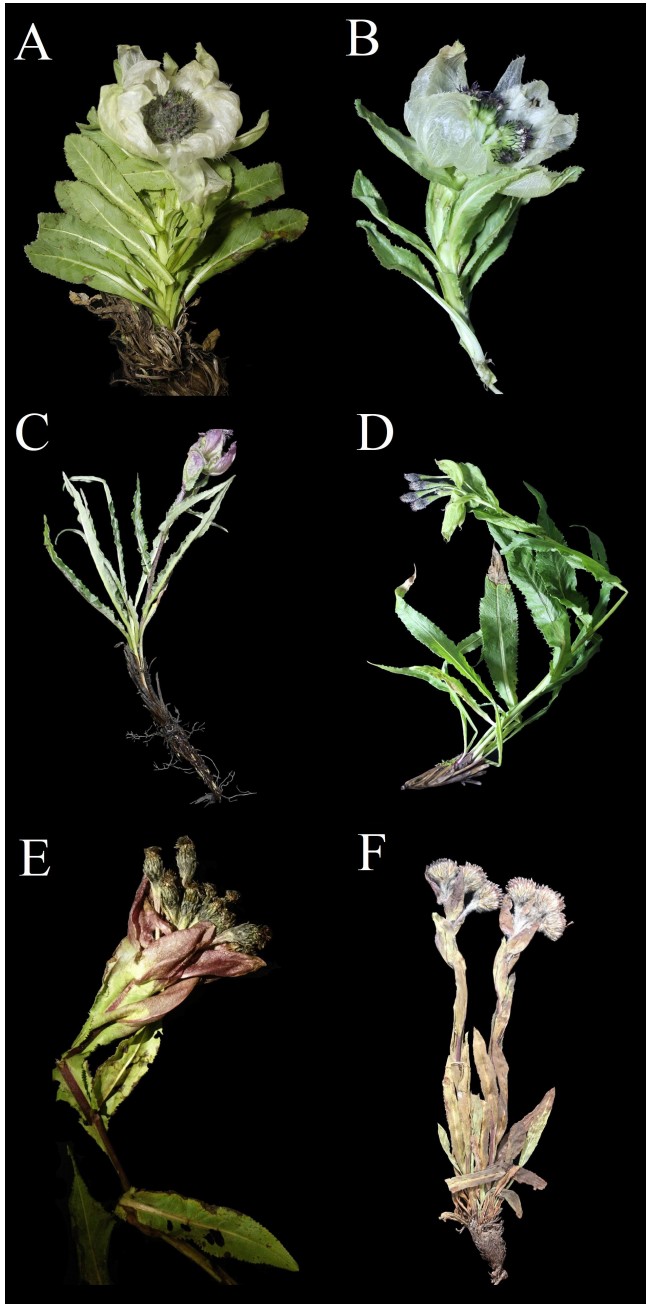

**Figure 1** **Photographs of six species sampled in the study.** (A) *S. bogedaensis*, WYJ201607018. (B) *S. involucrata*, WYJ201607025. (C) *S. pubifolia*, WYJ201607272. (D) *S. luae*, WYJ201607286. (E) *S. globosa*, WYJ201607422. (F) *S. erubescens*, sn110814017.

Previously, the sequences used in DNA barcodes for *Saussurea* species have been rather limited and only five species have been reported with DNA sequences. Among these species, none have been reported more than two populations, which is obviously insufficient for DNA barcode studies (*Wang et al., 2009*). Thus, in this study, we performed extensive

**Table 1  List of medicinal plants within *Saussurea* subg. *Amphilaena*.**

| Species | Reference |
| --- | --- |
| *S. involuvcrata* | *Chen, Pei & Zhao (2010)* and *Chik et al. (2015)* |
| *S. globosa* | *Cao et al. (2016)* and *Li (1999)* |
| *S. wettsteiniana* | *Jiang, Luo & Xu (2010)* |
| *S. polycolea* | *Jiang, Luo & Xu (2010)* and *Li (1999)* |
| *S. uniflora* | *Jiang, Luo & Xu (2010)* and *Li (1999)* |
| *S. velutina* | *Jiang, Luo & Xu (2010)* |
| *S. phaeantha* | *Cao et al. (2016)* and *Li (1999)* |
| *S. orgaadayi* | *Shi & Raab-Straube (2011)* |
| *S. tangutica* | *Cao et al. (2016)* and *Li, Zhu & Cai (2000)* |
| *S. bracteata* | *Li (1999)* |
| *S. erubescens* | *Cao et al. (2016)* and *Li (1999)* |
| *S. nigrescens* | *Cao et al. (2016)* and *Li (1999)* |
| *S. iodostegia* | *Cao et al. (2016)* and *Li (1999)* |
| *S. glandulosissima* | *Cao et al. (2016)*, *Li (1999)* and *Yang et al. (2005)* |
| *S. sikkimensis* | *Cao et al. (2016)*, *Li (1999)* and *Yang et al. (2005)* |

investigations in the field, and we sequenced five DNA barcode candidates in chloroplasts (*mat*K, *trn*H-*psb*A, *trn*K, and *rbc*L) and the nuclear ITS. Our main aims were: (i) to evaluate the application of these DNA barcodes in *S.* subg. *Amphilaena*; (ii) to develop an objective method for identifying medically important *Saussurea* species; and (iii) to explore the possible taxonomic problems and potential substitutes for some rare herbs.

# MATERIALS AND METHODS

## Taxon sampling

In total, 20 species were sampled in the present study, including 18 from the 38 species recognized in the latest monograph on *S.* subg. *Amphilaena* (*Raab-Straube, 2017*), one recently published species, *S. bogedaensis* (*Chen & Wang, 2018*), and a *Jurinea* species, which was selected as an outgroup according to a previous study (*Wang et al., 2009*). Photos of some species are presented in Fig. 1. Our sample focus on medical resources and 15 species formally recorded in the medical literature were included in the analyses (Table 1). For most of the species in the ingroup, we collected from two or more populations, with more than three individuals from each population. In total, we collected 132 individuals and their details are listed in Table 2.

## DNA extraction, PCR amplification, and sequencing

Genomic DNA was extracted from dried leaves in silica gel using the CTAB method (*Doyle, 1987*). Five regions (*rbc*L, *mat*K, *trn*H-*psb*A, *trn*K, and ITS) (*Berends, Jones & Mullet, 1990*; *Ford et al., 2009*; *Olmstead et al., 1992*; *Sang, Crawford & Stuessy, 1997*; *White et al., 1990*), were amplified and sequenced using the primers listed in Table 3. A PCR reaction mixture comprising 25 μL was prepared and amplified according to the procedure described by *Wang et al. (2009)*. The PCR products were sent to the Beijing Genomics Institute for

Chen et al. (2019), *PeerJ*, DOI 10.7717/peerj.6357

**Table 2** The name, locality, voucher and GenBank accession number for the samples used in this study.

| Species | Locality (All from China) | Voucher/Individual | Latitude (°) | Longitude (°) | Altitude (m) | GenBank accession number (ITS, *mat*K, *rbc*L, *trn*K, *trn*H-*psb*A) | | | | |
|---|---|---|---|---|---|---|---|---|---|---|
| S. bogedaensis | Qitai, Xinjiang | WYJ201607018b, 140 | 43.45321 | 89.55213 | 3,471 | MH003705 | MH070617 | MH070870 | MH070996 | MH070743 |
| S. bogedaensis | Qitai, Xinjiang | WYJ201607018a, 167 | 43.45321 | 89.55213 | 3,471 | MH003706 | MH070618 | MH070871 | MH070997 | MH070744 |
| S. bogedaensis | Qitai, Xinjiang | WYJ201607018, 378 | 43.45321 | 89.55213 | 3,471 | MH003707 | MH070619 | MH070872 | MH070998 | MH070745 |
| S. bogedaensis | Qitai, Xinjiang | WYJ201308006, 38 | 43.44370 | 89.58167 | 3,386 | MH003708 | MH070620 | MH070873 | MH070999 | MH070746 |
| S. bogedaensis | Qitai, Xinjiang | WYJ201308006, 39 | 43.44370 | 89.58167 | 3,386 | MH003709 | MH070621 | MH070874 | MH071000 | MH070747 |
| S. bogedaensis | Qitai, Xinjiang | WYJ201308006, 40 | 43.44370 | 89.58167 | 3,386 | MH003710 | MH070622 | MH070875 | MH071001 | MH070748 |
| S. bracteata | Qumalai, Qinghai | WYJ201207537, 114 | 34.84716 | 94.94569 | 4,621 | MH003711 | MH070623 | MH070876 | MH071002 | MH070749 |
| S. bracteata | Cuomei, Xizang | WYJ201607213, 151 | 28.51474 | 91.45611 | 4,934 | MH003712 | MH070624 | MH070877 | MH071003 | MH070750 |
| S. bracteata | Cuomei, Xizang | WYJ201607213, 153 | 28.51474 | 91.45611 | 4,934 | MH003713 | MH070625 | MH070878 | MH071004 | MH070751 |
| S. bracteata | Yushu, Qinghai | WYJ201607043, 160 | 35.05681 | 93.01225 | 4,644 | MH003714 | MH070626 | MH070879 | MH071005 | MH070752 |
| S. bracteata | Yushu, Qinghai | WYJ201607043, 161 | 35.05681 | 93.01225 | 4,644 | MH003715 | MH070627 | MH070880 | MH071006 | MH070753 |
| S. bracteata | Yushu, Qinghai | WYJ201607043, 162 | 35.05681 | 93.01225 | 4,644 | MH003716 | MH070628 | MH070881 | MH071007 | MH070754 |
| S. bracteata | Jilong, Xizang | WYJ201607099, 173 | 28.93494 | 85.39376 | 5,108 | MH003717 | MH070629 | MH070882 | MH071008 | MH070755 |
| S. bracteata | Jilong, Xizang | WYJ201607099, 174 | 28.93494 | 85.39376 | 5,108 | MH003718 | MH070630 | MH070883 | MH071009 | MH070756 |
| S. bracteata | Jilong, Xizang | WYJ201607099, 175 | 28.93494 | 85.39376 | 5,108 | MH003719 | MH070631 | MH070884 | MH071010 | MH070757 |
| S. bracteata | Geermu, Qinghai | WYJ201607053f, 204 | 32.98834 | 91.98589 | 5,120 | MH003720 | MH070632 | MH070885 | MH071011 | MH070758 |
| S. bracteata | Geermu, Qinghai | WYJ201607041, 248 | 35.51127 | 93.72552 | 4,525 | MH003721 | MH070633 | MH070886 | MH071012 | MH070759 |
| S. bracteata | Geermu, Qinghai | WYJ201607041, 249 | 35.51127 | 93.72552 | 4,525 | MH003722 | MH070634 | MH070887 | MH071013 | MH070760 |
| S. erubescens | Luqu, Gansu | sn110814017, 123 | 34.59103 | 102.48699 | 3,345 | MH003723 | MH070635 | MH070888 | MH071014 | MH070761 |
| S. erubescens | Luqu, Gansu | sn110814018, 124 | 34.59121 | 102.48657 | 3,367 | MH003724 | MH070636 | MH070889 | MH071015 | MH070762 |
| S. erubescens | Luqu, Gansu | sn110814017, 353 | 34.59103 | 102.48699 | 3,345 | MH003725 | MH070637 | MH070890 | MH071016 | MH070763 |
| S. erubescens | Luqu, Gansu | sn110815020, 355 | 33.59203 | 101.48659 | 3,451 | MH003726 | MH070638 | MH070891 | MH071017 | MH070764 |
| S. erubescens | Xiahe, Gansu | Ikeda200713210, 371 | 35.20252 | 102.52181 | 3,342 | MH003727 | MH070639 | MH070892 | MH071018 | MH070765 |
| S. globosa | Aba, Sicuan | WYJ-2011-175, 109 | 33.63526 | 102.35556 | 3,470 | MH003728 | MH070640 | MH070893 | MH071019 | MH070766 |
| S. globosa | Baoxing, Sicuan | WYJ201607422, 168 | 30.49153 | 102.48188 | 3,992 | MH003729 | MH070641 | MH070894 | MH071020 | MH070767 |
| S. globosa | Kangding, Sicuan | WYJ201209151, 318 | 30.05441 | 101.96308 | 3,841 | MH003730 | MH070642 | MH070895 | MH071021 | MH070768 |
| S. globosa | Kangding, Sicuan | WYJ201209158, 329 | 30.05564 | 101.97304 | 3,864 | MH003731 | MH070643 | MH070896 | MH071022 | MH070769 |
| S. globosa | Kangding, Sicuan | WYJ201209157, 331 | 30.13242 | 101.56306 | 3,974 | MH003732 | MH070644 | MH070897 | MH071023 | MH070770 |
| S. globosa | – | – | – | – | – | EF420926 | – | – | – | – |
| S. globosa | Xiangcheng, Sicuan | WYJ201209234, 337 | 28.93118 | 99.79842 | 3,764 | MH003733 | – | – | – | – |
| S. globosa | Xiangcheng, Sicuan | WYJ-2011-069, 80 | 28.53118 | 99.45658 | 3,835 | MH003734 | MH070645 | MH070898 | MH071024 | MH070771 |
| S. globosa | Xiangcheng, Sicuan | WYJ-2011-069, 81 | 28.53118 | 99.45658 | 3,835 | MH003735 | MH070646 | MH070899 | MH071025 | MH070772 |
| S. involucrata | Urumqi, Xinjiang | WYJ201607025a, 163 | 43.10847 | 86.84220 | 3,564 | MH003736 | MH070647 | MH070900 | MH071026 | MH070773 |
| S. involucrata | Urumqi, Xinjiang | WYJ201607025c, 165 | 43.10847 | 86.84220 | 3,564 | MH003737 | MH070648 | MH070901 | MH071027 | MH070774 |

Chen et al. (2019), *PeerJ*, DOI 10.7717/peerj.6357

| Species | Locality (All from China) | Voucher/Individual | Latitude (°) | Longitude (°) | Altitude (m) | GenBank accession number (ITS, *mat*K, *rbc*L, *trn*K, *trn*H-*psb*A) | | | | |
|---------|---------------------------|--------------------|--------------|---------------|--------------|------|------|------|------|------|
| *S. involucrata* | Tekesi, Xinjiang | WYJ201308184, 24 | 43.09915 | 82.68382 | 3,678 | MH003738 | MH070649 | MH070902 | MH071028 | MH070775 |
| *S. involucrata* | Tekesi, Xinjiang | WYJ201308184, 26 | 43.09915 | 82.68382 | 3,678 | MH003739 | MH070650 | MH070903 | MH071029 | MH070776 |
| *S. involucrata* | Urumqi, Xinjiang | WYJ201308203, 372 | 43.11985 | 86.82125 | 3,768 | MH003740 | MH070651 | MH070904 | MH071030 | MH070777 |
| *S. involucrata* | Urumqi, Xinjiang | WYJ201308203, 374 | 43.11985 | 86.82125 | 3,768 | MH003741 | MH070652 | MH070905 | MH071031 | MH070778 |
| *S. involucrata* | Xinyuan, Xinjiang | WYJ201308188, 390 | 43.33469 | 84.01032 | 3,543 | MH003742 | MH070653 | MH070906 | MH071032 | MH070779 |
| *S. involucrata* | Urumqi, Xinjiang | WYJ201308203, 41 | 43.11985 | 86.82125 | 3,768 | MH003743 | MH070654 | MH070907 | MH071033 | MH070780 |
| *S. involucrata* | Xinyuan, Xinjiang | WYJ201308188, 47 | 43.33469 | 84.01032 | 3,543 | MH003744 | MH070655 | MH070908 | MH071034 | MH070781 |
| *S. involucrata* | Xinyuan, Xinjiang | WYJ201308188, 48 | 43.33469 | 84.01032 | 3,543 | MH003745 | MH070656 | MH070909 | MH071035 | MH070782 |
| *S. involucrata* | Dushanzi, Xinjiang | WYJ201308131, 61 | 43.77545 | 84.45615 | 2,684 | MH003746 | MH070657 | MH070910 | MH071036 | MH070783 |
| *S. involucrata* | Dushanzi, Xinjiang | WYJ201308131, 63 | 43.77545 | 84.45615 | 2,684 | MH003747 | MH070658 | MH070911 | MH071037 | MH070784 |
| *S. iodostegia* | Datong, Shanxi | WYJ201507117, 107 | 39.05578 | 113.65927 | 2,514 | MH003748 | MH070659 | MH070912 | MH071038 | MH070785 |
| *S. iodostegia* | Datong, Shanxi | WYJ201507117, 108 | 39.05578 | 113.65927 | 2,514 | MH003749 | MH070660 | MH070913 | MH071039 | MH070786 |
| *S. iodostegia* | Weixian, Hebei | WYJ201309004, 20 | 39.91413 | 114.96546 | 2,237 | MH003750 | MH070661 | MH070914 | MH071040 | MH070787 |
| *S. iodostegia* | Weixian, Hebei | WYJ201309004, 21 | 39.91413 | 114.96546 | 2,237 | MH003751 | MH070662 | MH070915 | MH071041 | MH070788 |
| *S. iodostegia* | Weixian, Hebei | WYJ201309004, 22 | 39.91413 | 114.96546 | 2,237 | MH003752 | MH070663 | MH070916 | MH071042 | MH070789 |
| *S. iodostegia* | Mentougou, Beijing | WYJ201507105, 27 | 40.03633 | 115.47206 | 2,048 | MH003753 | MH070664 | MH070917 | MH071043 | MH070790 |
| *S. iodostegia* | Mentougou, Beijing | WYJ201507105, 28 | 40.03633 | 115.47206 | 2,048 | MH003754 | MH070665 | MH070918 | MH071044 | MH070791 |
| *S. iodostegia* | Mentougou, Beijing | WYJ201507105, 29 | 40.03633 | 115.47206 | 2,048 | MH003755 | MH070666 | MH070919 | MH071045 | MH070792 |
| *S. luae* | Linzhi, Xizang | WYJ201607286a, 271 | 29.59022 | 94.59631 | 4,121 | MH003756 | – | – | – | – |
| *S. luae* | Linzhi, Xizang | WYJ201607286a, 272 | 29.59022 | 94.59631 | 4,121 | MH003757 | – | – | – | – |
| *S. luae* | Linzhi, Xizang | WYJ201607286b, 273 | 29.59022 | 94.59631 | 4,121 | MH003758 | MH070667 | MH070920 | MH071046 | MH070793 |
| *S. luae* | Linzhi, Xizang | WYJ201607286c, 283 | 29.59022 | 94.59631 | 4,121 | MH003759 | – | – | – | – |
| *S. luae* | Linzhi, Xizang | LJQ2620, 316 | 28.48051 | 93.36541 | 4,225 | MH003760 | MH070668 | MH070921 | MH071047 | MH070794 |
| *S. nigrescens* | Tianzhu, Gansu | LJQ1480, 314 | 36.41075 | 102.45620 | 1,900 | MH003761 | MH070669 | MH070922 | MH071048 | MH070795 |
| *S. nigrescens* | Sunan, Gansu | LJQ1517, 315 | 37.23345 | 102.32444 | 2,651 | MH003762 | MH070670 | MH070923 | MH071049 | MH070796 |
| *S. nigrescens* | Huangyuan, Qinghai | Liu1603, 320 | 36.20387 | 98.14870 | 3,700 | MH003763 | MH070671 | MH070924 | MH071050 | MH070797 |
| *S. nigrescens* | Huangzhong, Qinghai | WYJ200611, 347 | 36.50087 | 101.57164 | 3,641 | MH003764 | MH070672 | MH070925 | MH071051 | MH070798 |
| *S. nigrescens* | Menyuan, Qinghai | LJQ-QLS-2008-0065, 82 | 37.37502 | 101.62422 | 2,654 | MH003765 | MH070673 | MH070926 | MH071052 | MH070799 |
| *S. nigrescens* | Menyuan, Qinghai | LJQ-QLS-2008-0065, 83 | 37.37502 | 101.62422 | 2,654 | MH003766 | MH070674 | MH070927 | MH071053 | MH070800 |
| *S. nigrescens* | Menyuan, Qinghai | LJQ-QLS-2008-0065, 84 | 37.37502 | 101.62422 | 2,654 | MH003767 | MH070675 | MH070928 | MH071054 | MH070801 |
| *S. glandulosissima* | Chayu, Xizang | WYJ201607321, 257 | 29.32542 | 97.134728 | 3,949 | MH003768 | MH070676 | MH070929 | MH071055 | MH070802 |
| *S. glandulosissima* | Linzhi, Xizang | WYJ201607298, 264 | 29.627012 | 94.635744 | 4,433 | MH003769 | MH070677 | MH070930 | MH071056 | MH070803 |
| *S. glandulosissima* | Linzhi, Xizang | WYJ201607298, 379 | 29.627012 | 94.635744 | 4,433 | MH003770 | MH070678 | MH070931 | MH071057 | MH070804 |
| *S. glandulosissima* | Chayu, Xizang | WYJ201607321, 382 | 29.32542 | 97.134728 | 3,949 | MH003771 | MH070679 | MH070932 | MH071058 | MH070805 |
| *S. glandulosissima* | Chayu, Xizang | WYJ201607321, 383 | 29.32542 | 97.134728 | 3,949 | MH003772 | MH070680 | MH070933 | MH071059 | MH070806 |
| *S. orgaadayi* | Altay, Xinjiang | WYJ201308041, 11 | 47.21846 | 89.87999 | 3,541 | MH003773 | MH070681 | MH070934 | MH071060 | MH070807 |

**Table 2** (*continued*)

| Species | Locality (All from China) | Voucher/Individual | Latitude (°) | Longitude (°) | Altitude (m) | GenBank accession number (ITS, *mat*K, *rbc*L, *trn*K, *trn*H-*psb*A) | | | | |
|---|---|---|---|---|---|---|---|---|---|---|
| *S. orgaadayi* | Altay, Xinjiang | WYJ201308041, 12 | 47.21846 | 89.87999 | 3,541 | MH003774 | MH070682 | MH070935 | MH071061 | MH070808 |
| *S. orgaadayi* | Altay, Xinjiang | WYJ201308041, 360 | 47.21846 | 89.87999 | 3,541 | MH003775 | MH070683 | MH070936 | MH071062 | MH070809 |
| *S. phaeantha* | Xiaojing, Sicuan | WYJ201209126, 1 | 30.99918 | 102.3644 | 3,642 | MH003776 | MH070684 | MH070937 | MH071063 | MH070810 |
| *S. phaeantha* | Xiaojing, Sicuan | WYJ201209126, 2 | 30.99918 | 102.3644 | 3,642 | MH003779 | MH070687 | MH070940 | MH071066 | MH070813 |
| *S. phaeantha* | Qilian, Gansu | WYJ201607014, 195 | 38.60685 | 99.48221 | 4,096 | MH003777 | MH070685 | MH070938 | MH071064 | MH070811 |
| *S. phaeantha* | Qilian, Gansu | WYJ201607014, 196 | 38.60685 | 99.48221 | 4,096 | MH003778 | MH070686 | MH070939 | MH071065 | MH070812 |
| *S. phaeantha* | Maqin, Qinghai | LJQ1718, 317 | 34.47733 | 100.23956 | 3,210 | MH003780 | MH070688 | MH070941 | MH071067 | MH070814 |
| *S. phaeantha* | Xinghai, Qinghai | sn110718001, 349 | 35.58868 | 99.98818 | 2,654 | MH003781 | MH070689 | MH070942 | MH071068 | MH070815 |
| *S. phaeantha* | Xinghai, Qinghai | sn120811001, 351 | 34.32412 | 99.35641 | 2,641 | MH003782 | MH070690 | MH070943 | MH071069 | MH070816 |
| *S. phaeantha* | Xinghai, Qinghai | sn120801130, 354 | 35.38821 | 99.78935 | 2,684 | MH003783 | – | – | – | MH070817 |
| *S. polycolea* | Linzhi, Xizang | WYJ201607292, 229 | 29.62701 | 94.63574 | 4,433 | MH003784 | MH070691 | MH070944 | MH071070 | MH070818 |
| *S. polycolea* | Linzhi, Xizang | WYJ201607292, 230 | 29.62701 | 94.63574 | 4,433 | MH003785 | MH070692 | MH070945 | MH071071 | MH070819 |
| *S. polycolea* | Linzhi, Xizang | WYJ201607292, 231 | 29.62701 | 94.63574 | 4,433 | MH003786 | MH070693 | MH070946 | MH071072 | MH070820 |
| *S. polycolea* | Langxian, Xizang | WYJ201607279, 269 | 28.883036 | 93.356181 | 4,472 | MH003787 | MH070694 | MH070947 | MH071073 | MH070821 |
| *S. polycolea* | Langxian, Xizang | WYJ201607279, 270 | 28.883036 | 93.356181 | 4,472 | MH003788 | MH070695 | MH070948 | MH071074 | MH070822 |
| *S. polycolea* | Linzhi, Xizang | Liu07257, 334 | 29.62201 | 94.63554 | 4,231 | MH003789 | MH070696 | MH070949 | MH071075 | MH070823 |
| *S. pubifolia* | Jiacha, Xizang | WYJ201607272a, 206 | 29.03175 | 92.35724 | 4,796 | MH003790 | MH070697 | MH070950 | MH071076 | MH070824 |
| *S. pubifolia* | Jiacha, Xizang | WYJ201607272b, 207 | 29.03175 | 92.35724 | 4,796 | MH003791 | MH070698 | MH070951 | MH071077 | MH070825 |
| *S. pubifolia* | Jiacha, Xizang | WYJ201607272c, 208 | 29.03175 | 92.35724 | 4,796 | MH003792 | MH070699 | MH070952 | MH071078 | MH070826 |
| *S. pubifolia* | Jiacha, Xizang | WYJ-2011-057, 94 | 29.02165 | 92.35714 | 4,786 | MH003793 | MH070700 | MH070953 | MH071079 | MH070827 |
| *S. sikkimensis* | Cuona, Xizang | WYJ201607242, 156 | 27.92057 | 91.84863 | 3,970 | MH003794 | MH070701 | MH070954 | MH071080 | MH070828 |
| *S. sikkimensis* | Yadong, Xizang | WYJ201607150e, 186 | 27.48592 | 88.90708 | 4,102 | MH003795 | MH070702 | MH070955 | MH071081 | MH070829 |
| *S. sikkimensis* | Yadong, Xizang | WYJ201607150c, 187 | 27.48592 | 88.90708 | 4,102 | MH003796 | MH070703 | MH070956 | MH071082 | MH070830 |
| *S. sikkimensis* | Yadong, Xizang | WYJ201607150f, 385 | 27.48592 | 88.90708 | 4,102 | MH003797 | MH070704 | MH070957 | MH071083 | MH070831 |
| *S. sikkimensis* | Yadong, Xizang | WYJ201607150 h, 386 | 27.48592 | 88.90708 | 4,102 | MH003798 | MH070705 | MH070958 | MH071084 | MH070832 |
| *S. sikkimensis* | Cuona, Xizang | WYJ201607242, 388 | 27.92057 | 91.84863 | 3,970 | MH003799 | MH070706 | MH070959 | MH071085 | MH070833 |
| *S. sikkimensis* | Cuona, Xizang | WYJ201607242, 389 | 27.92057 | 91.84863 | 3,970 | MH003800 | MH070707 | MH070960 | MH071086 | MH070834 |
| *S. tangutica* | Qilian, Gansu | WYJ201607013, 226 | 38.60685 | 99.48221 | 4,096 | MH003801 | MH070708 | MH070961 | MH071087 | MH070835 |
| *S. tangutica* | Qilian, Gansu | WYJ201607013, 228 | 38.60685 | 99.48221 | 4,096 | MH003802 | MH070709 | MH070962 | MH071088 | MH070836 |
| *S. tangutica* | Zhiduo, Qinghai | WYJ201207279, 328 | 33.85203 | 95.61335 | 3,948 | MH003803 | MH070710 | MH070963 | MH071089 | MH070837 |
| *S. tangutica* | Kangding, Sicuan | sn120801019, 332 | 30.05093 | 101.96437 | 3,987 | MH003804 | MH070711 | MH070964 | MH071090 | MH070838 |
| *S. tangutica* | Kangding, Sicuan | sn120801019, 335 | 30.05093 | 101.96437 | 3,987 | MH003805 | MH070712 | MH070965 | MH071091 | MH070839 |
| *S. tangutica* | Zhiduo, Qinghai | WYJ201207279, 340 | 33.85203 | 95.61335 | 3,948 | MH003806 | MH070713 | MH070966 | MH071092 | MH070840 |
| *S. uniflora* | Cuona, Xizang | WYJ201607254, 142 | 27.765831 | 91.90194 | 4,138 | MH003807 | MH070714 | MH070967 | MH071093 | MH070841 |
| *S. uniflora* | Cuona, Xizang | WYJ201607254, 143 | 27.765831 | 91.90194 | 4,138 | MH003808 | MH070715 | MH070968 | MH071094 | MH070842 |
| *S. uniflora* | Cuona, Xizang | WYJ201607254, 144 | 27.765831 | 91.90194 | 4,138 | MH003809 | MH070716 | MH070969 | MH071095 | MH070843 |

**Table 2** (*continued*)

| Species | Locality (All from China) | Voucher/Individual | Latitude (°) | Longitude (°) | Altitude (m) | GenBank accession number (ITS, *mat*K, *rbc*L, *trn*K, *trn*H-*psb*A) | | | | |
|---------|---------------------------|--------------------|--------------|----------------|--------------|---------|---------|---------|---------|---------|
| *S. uniflora* | Yadong, Xizang | WYJ201607151c, 145 | 27.48592 | 88.90708 | 4,102 | MH003810 | MH070717 | MH070970 | MH071096 | MH070844 |
| *S. uniflora* | Yadong, Xizang | WYJ201607151a, 146 | 27.48592 | 88.90708 | 4,102 | MH003811 | MH070718 | MH070971 | MH071097 | MH070845 |
| *S. uniflora* | Yadong, Xizang | WYJ201607151b, 147 | 27.48592 | 88.90708 | 4,102 | MH003812 | – | – | – | – |
| *S. uniflora* | Cuona, Xizang | WYJ201607243, 197 | 27.92057 | 91.84863 | 3,970 | MH003813 | MH070719 | MH070972 | MH071098 | MH070846 |
| *S. veitchiana* | Xinglong, Hebei | WYJ201507098, 302 | 40.59808 | 117.47655 | 2,032 | MH003814 | MH070720 | MH070973 | MH071099 | MH070847 |
| *S. veitchiana* | Xinglong, Hebei | WYJ201507098, 303 | 40.59808 | 117.47655 | 2,032 | MH003815 | MH070721 | MH070974 | MH071100 | MH070848 |
| *S. veitchiana* | Nuanchuan, Henan | WYJ201507135, 52 | 33.67057 | 111.79417 | 1,651 | MH003816 | MH070722 | MH070975 | MH071101 | MH070849 |
| *S. veitchiana* | Nuanchuan, Henan | WYJ201507135, 53 | 33.67057 | 111.79417 | 1,651 | MH003817 | MH070723 | MH070976 | MH071102 | MH070850 |
| *S. veitchiana* | Nuanchuan, Henan | WYJ201507135, 54 | 33.67057 | 111.79417 | 1,651 | MH003818 | MH070724 | MH070977 | MH071103 | MH070851 |
| *S. veitchiana* | Nuanchuan, Henan | WYJ201507135, 55 | 33.67057 | 111.79417 | 1,651 | MH003819 | MH070725 | MH070978 | MH071104 | MH070852 |
| *S. veitchiana* | Shenlongjia, Hubei | WYJ201507160, 57 | 31.43997 | 110.307149 | 3,098 | MH003820 | MH070726 | MH070979 | MH071105 | MH070853 |
| *S. veitchiana* | Shenlongjia, Hubei | WYJ201507160, 58 | 31.43997 | 110.307149 | 3,098 | MH003821 | MH070727 | MH070980 | MH071106 | MH070854 |
| *S. veitchiana* | Shenlongjia, Hubei | WYJ201507160, 59 | 31.43997 | 110.307149 | 3,098 | MH003822 | MH070728 | MH070981 | MH071107 | MH070855 |
| *S. veitchiana* | Wuxi, Chongqing | WYJ201507184, 64 | 31.43791 | 109.15498 | 1,795 | MH003823 | MH070729 | MH070982 | MH071108 | MH070856 |
| *S. veitchiana* | Wuxi, Chongqing | WYJ201507184, 65 | 31.43791 | 109.15498 | 1,795 | MH003824 | MH070730 | MH070983 | MH071109 | MH070857 |
| *S. veitchiana* | Wuxi, Chongqing | WYJ201507184, 66 | 31.43791 | 109.15498 | 1,795 | MH003825 | MH070731 | MH070984 | MH071110 | MH070858 |
| *S. veitchiana* | Wuxi, Chongqing | WYJ201507184, 67 | 31.43791 | 109.15498 | 1,795 | MH003826 | MH070732 | MH070985 | MH071111 | MH070859 |
| *S. velutina* | Xiaojin, Sichuan | WYJ201209124, 339 | 30.99441 | 102.82915 | 4,000 | MH003827 | MH070733 | MH070986 | MH071112 | MH070860 |
| *S. velutina* | Xiaojin, Sichuan | WYJ201209124, 342 | 30.99441 | 102.82915 | 4,000 | MH003828 | MH070734 | MH070987 | MH071113 | MH070861 |
| *S. velutina* | Xiaojin, Sichuan | WYJ201209124, 76 | 30.99441 | 102.82915 | 4,000 | MH003829 | MH070735 | MH070988 | MH071114 | MH070862 |
| *S. velutina* | Xiaojin, Sichuan | WYJ201209124, 77 | 30.99441 | 102.82915 | 4,000 | MH003830 | MH070736 | MH070989 | MH071115 | MH070863 |
| *S. velutina* | Xiaojin, Sichuan | WYJ201209124, 78 | 30.99441 | 102.82915 | 4,000 | MH003831 | MH070737 | MH070990 | MH071116 | MH070864 |
| *S. wettsteiniana* | Mianning, Sichuan | WYJ201607408a, 176 | 29.00106 | 102.14985 | 3,381 | MH003832 | MH070738 | MH070991 | MH071117 | MH070865 |
| *S. wettsteiniana* | Mianning, Sichuan | WYJ201607408b, 177 | 29.00106 | 102.14985 | 3,381 | MH003833 | MH070739 | MH070992 | MH071118 | MH070866 |
| *S. wettsteiniana* | Mianning, Sichuan | WYJ201607402, 178 | 29.00106 | 102.14985 | 3,381 | MH003834 | MH070740 | MH070993 | MH071119 | MH070867 |
| *S. wettsteiniana* | Mianning, Sichuan | WYJ201607402, 284 | 29.00106 | 102.14985 | 3,381 | MH003835 | MH070741 | MH070994 | MH071120 | MH070868 |
| *Jurinea multiflora* | Tuoli, Xinjiang | WYJ201308102, 377 | 45.73564 | 83.14712 | 1,753 | MH003704 | MH070616 | MH070869 | MH070995 | MH070742 |

**Table 3** List of the primers used in this study.

| Primer | Fragment | Sequence(5′–3′) | Reference |
| --- | --- | --- | --- |
| ITS4 | ITS | TCCTCCGCTTATTGATATGC | *White et al. (1990)* |
| ITS1 | ITS | AGAAGTCGTAACAAGGTTTCCGTAGG | *White et al. (1990)* |
| *trn*K(UUU) | *trn*K | TTAAAAGCCGAGTACTCTACC | *Berends, Jones & Mullet (1990)* |
| *rps*16 | *trn*K | AAAGTGGGTTTTTATGATCC | *Berends, Jones & Mullet (1990)* |
| *psb*A | *psb*A | GTTATGCATGAACGTAATGCTC | *Sang, Crawford & Stuessy (1997)* |
| *trn*H | *psb*A | CGCGCATGGTGGATTCACAATCC | *Sang, Crawford & Stuessy (1997)* |
| *mat*K-xf | *mat*K | TAATTTACGATCAATTCATTC | *Ford et al. (2009)* |
| *mat*K-5r | *mat*K | GTTCTAGCACAAGAAAGTCG | *Ford et al. (2009)* |
| *rbc*L1 | *rbc*L | ATGTCACCACAAACAGAGACTAAAGC | *Olmstead et al. (1992)* |
| *rbc*L911 | *rbc*L | TTTCTTCGCATGTACCCGC | *Olmstead et al. (1992)* |

commercial sequencing. Sequences were aligned using CLUSTALX v.2.1 (*Thompson et al., 1997*) with the default settings and adjusted manually with Bioedit v.7.0.5 (*Hall, 1999*). All of the sequences were registered in GenBank (Table 2).

## Data analysis

We constructed 31 datasets for ITS, *psb*A-*trn* H, *mat*K, and *trn*K, either individually or in different combinations. For the combination of ITS and each chloroplast loci, incongruence length difference (ILD) was preferred to test the incongruence (*Farris et al., 1995*) using PAUP version 4b10 (*Swofford, 2003*). For each dataset, the inter- and intraspecific genetic divergences were calculated as described by *Meyer & Paulay (2005)* and used to determine whether a barcoding gap was present. For each dataset, best close match (BCM) and two tree-based methods comprising neighbor-joining (NJ) and Bayesian inference (BI) were employed to analyze the five single markers and their different combinations. BCM analysis was conducted using the SPIDER package in R (*Brown et al., 2012*). NJ trees were constructed using PAUP with the Kimura two-parameter model (*Swofford, 2003*). Support for nodes was assessed based on 100,000 bootstrap replicates. BI analysis was implemented using MrBayes on XSEDE (v3.2.6) (*Ronquist et al., 2012*) and the optimal models for each marker were determined according to Akaike's information criterion with jModelTest2 in XSEDE (v2.1.6) (*Darriba et al., 2012*). Species were considered to be identified successfully if individual samples of a species clustered in species-specific monophyletic clades.

## RESULTS

The PCR amplification ranged from about 73% (*trn*K) to 93% (ITS), while sequencing success rates from about 95% for the three chloroplast loci to 100% for the ITS, as shown in Table 4. The length after alignment, the variable sites, the interspecific or intraspecific genetic distance for each locus as well as the *p* values of ILD test between ITS and each chloroplast locus are also listed in Table 4. The mean intraspecific genetic distances for each species based on ITS and the four cp markers combined are listed in Table 5, and those for the mean interspecific genetic distances are shown in Table 6. The distributions of the intraspecific and interspecific distances for each species based on the five separate
**Table 4** List of statistics information of five DNA barcodes and the result of incongruence length difference (ILD) analysis between ITS and each chloroplast locus.

| DNA region | ITS | *trn*H-*psb*A | *mat*K | *rbc*L | *trn*K |
|---|---|---|---|---|---|
| PCR success (%) | 92.7 | 77 | 89.6 | 91.6 | 72.9 |
| Sequencing success (%) | 100 | 96.18 | 95.42 | 95.42 | 95.42 |
| Aligned sequence length (bp) | 656 | 444 | 711 | 634 | 656 |
| No. indel (length in bp) | 3 (1) | 5 (1–3) | 0 | 0 | 4 (1) |
| No. variated sites | 111 | 22 | 18 | 8 | 28 |
| No. sampled species (individual) | 19 (131) | 19 (131) | 19 (131) | 19 (131) | 19 (131) |
| Interspecific distance mean (range) (%) | 0.011 (0-0.028) | 0.004(0–0.028) | 0.003(0–0.008) | 0.002(0–0.006) | 0.004(0–0.012) |
| Intraspecific distance mean (range) (%) | 0.001(0–0.005) | 0.002(0–0.021) | 0.001(0–0.006) | 0.001(0–0.006) | 0.001(0–0.009) |
| *p* values of ILD test between ITS | – | 0.02 | 0.001 | 0.12 | 0.001 |

**Table 5** Mean intraspecies distance (%) of ITS and the combined sequences of four chloroplast loci for each species.

| Species | ITS | Chloroplast |
|---|---|---|
| *S. bogedaensis* | 0.0 | 0.02 |
| *S. bracteata* | 0.0 | 0.00 |
| *S. erubescens* | 0.0 | 0.00 |
| *S. glandulosissima* | 0.1 | 0.07 |
| *S. globosa* | 0.2 | 0.04 |
| *S. involucrata* | 0.2 | 0.06 |
| *S. iodostegia* | 0.0 | 0.05 |
| *S. luae* | 0.0 | 0.29 |
| *S. nigrescens* | 0.0 | 0.00 |
| *S. orgaadayi* | 0.0 | 0.00 |
| *S. phaeantha* | 0.4 | 0.04 |
| *S. polycolea* | 0.0 | 0.07 |
| *S. pubifolia* | 0.0 | 0.00 |
| *S. sikkimensis* | 0.2 | 0.06 |
| *S. tangutica* | 0.1 | 0.46 |
| *S. uniflora* | 0.1 | 0.15 |
| *S. veitchiana* | 0.1 | 0.39 |
| *S. velutina* | 0.0 | 0.21 |
| *S. wettsteiniana* | 0.0 | 0.00 |

markers are shown in Fig. 2. In general, the mean interspecific distances were higher than the intraspecific distances for the five markers. However, the ranges of the intra- and interspecific distances overlapped for all the barcodes tested in this study.

The discriminatory powers of all the loci both individually and in different combinations based on the three methods are listed in Table 7 (Figs. S1–S59). In general, BCM achieved higher success rates, followed by NJ and BI, but there were a few exceptions. Among the results obtained with a single barcode, ITS (84.2–93.2%) had the highest species

Table 6 **The pairwise distances (%) of ITS (lower left) and the combined chloroplast loci (upper right) from 19 species of *Saussurea*.** (1) *S. bogedaensis*, (2) *S. bracteata*, (3) *S. erubescens*, (4) *S. globosa*, (5) *S. involucrate*, (6) *S. iodostegia*, (7) *S. luae*, (8) *S. nigrescens*, (9) *S. glandulosissima*, (10) *S. orgaadayi*, (11) *S. phaeantha*, (12) *S. polycolea*, (13) *S. pubifolia*, (14) *S. sikkimensis*, (15) *S. tangutica*, (16) *S. uniflora*, (17) *S. veitchiana*, (18) *S. velutina*, (19) *S. wettsteiniana*.

| CP ITS | 1 | 2 | 3 | 4 | 5 | 6 | 7 | 8 | 9 | 10 | 11 | 12 | 13 | 14 | 15 | 16 | 17 | 18 | 19 |
|---|---|---|---|---|---|---|---|---|---|---|---|---|---|---|---|---|---|---|---|
| 1 | | 0.30 | 0.26 | 0.28 | 0.22 | 0.62 | 0.32 | 0.34 | 0.28 | 0.22 | 0.28 | 0.34 | 0.30 | 0.41 | 0.46 | 0.34 | 0.55 | 0.34 | 0.26 |
| 2 | 1.92 | | 0.04 | 0.06 | 0.17 | 0.57 | 0.19 | 0.29 | 0.22 | 0.16 | 0.06 | 0.12 | 0.00 | 0.35 | 0.35 | 0.23 | 0.50 | 0.16 | 0.21 |
| 3 | 1.52 | 2.77 | | 0.02 | 0.13 | 0.53 | 0.14 | 0.25 | 0.18 | 0.12 | 0.02 | 0.08 | 0.04 | 0.31 | 0.31 | 0.19 | 0.46 | 0.12 | 0.16 |
| 4 | 1.53 | 2.88 | 0.61 | | 0.15 | 0.55 | 0.17 | 0.27 | 0.20 | 0.15 | 0.05 | 0.10 | 0.06 | 0.34 | 0.33 | 0.22 | 0.48 | 0.15 | 0.19 |
| 5 | 0.93 | 2.58 | 2.14 | 2.14 | | 0.48 | 0.19 | 0.21 | 0.14 | 0.09 | 0.15 | 0.20 | 0.17 | 0.27 | 0.33 | 0.21 | 0.42 | 0.20 | 0.13 |
| 6 | 1.96 | 3.33 | 1.85 | 1.60 | 2.47 | | 0.59 | 0.53 | 0.54 | 0.49 | 0.55 | 0.60 | 0.57 | 0.51 | 0.71 | 0.55 | 0.37 | 0.57 | 0.53 |
| 7 | 1.07 | 0.72 | 1.90 | 1.78 | 1.72 | 2.31 | | 0.31 | 0.18 | 0.19 | 0.17 | 0.21 | 0.19 | 0.37 | 0.39 | 0.25 | 0.52 | 0.23 | 0.23 |
| 8 | 1.83 | 3.19 | 1.72 | 1.47 | 2.34 | 0.34 | 2.12 | | 0.26 | 0.21 | 0.27 | 0.32 | 0.29 | 0.31 | 0.45 | 0.22 | 0.32 | 0.19 | 0.25 |
| 9 | 1.35 | 2.69 | 1.56 | 1.31 | 1.92 | 1.74 | 1.69 | 1.60 | | 0.14 | 0.20 | 0.24 | 0.22 | 0.33 | 0.34 | 0.22 | 0.47 | 0.26 | 0.18 |
| 10 | 1.41 | 3.08 | 2.30 | 2.35 | 2.02 | 2.28 | 2.21 | 2.17 | 2.16 | | 0.15 | 0.20 | 0.16 | 0.27 | 0.32 | 0.21 | 0.42 | 0.20 | 0.12 |
| 11 | 1.53 | 2.84 | 1.60 | 1.45 | 2.14 | 1.92 | 1.84 | 1.78 | 1.31 | 2.34 | | 0.10 | 0.06 | 0.34 | 0.33 | 0.22 | 0.48 | 0.15 | 0.19 |
| 12 | 1.09 | 2.42 | 1.36 | 1.06 | 1.69 | 1.48 | 1.43 | 1.35 | 0.87 | 1.89 | 0.89 | | 0.12 | 0.37 | 0.37 | 0.26 | 0.53 | 0.20 | 0.24 |
| 13 | 1.61 | 1.32 | 2.22 | 2.23 | 2.26 | 3.00 | 0.23 | 2.84 | 2.37 | 2.76 | 2.51 | 2.10 | | 0.35 | 0.35 | 0.23 | 0.50 | 0.16 | 0.21 |
| 14 | 1.11 | 2.44 | 1.34 | 1.08 | 1.71 | 1.49 | 1.38 | 1.36 | 0.71 | 1.91 | 1.07 | 0.64 | 2.12 | | 0.51 | 0.34 | 0.48 | 0.35 | 0.31 |
| 15 | 1.63 | 2.98 | 1.58 | 1.59 | 1.47 | 2.57 | 2.01 | 2.42 | 2.06 | 2.67 | 2.20 | 1.78 | 2.32 | 1.81 | | 0.42 | 0.65 | 0.40 | 0.35 |
| 16 | 1.00 | 2.33 | 1.27 | 0.97 | 1.44 | 1.38 | 1.34 | 1.26 | 0.78 | 1.80 | 0.96 | 0.53 | 2.01 | 0.55 | 1.70 | | 0.46 | 0.24 | 0.25 |
| 17 | 2.10 | 3.48 | 2.06 | 1.74 | 2.62 | 1.52 | 2.36 | 1.30 | 1.72 | 2.93 | 2.02 | 1.62 | 2.81 | 1.64 | 2.50 | 1.53 | | 0.45 | 0.46 |
| 18 | 2.21 | 2.91 | 2.49 | 2.50 | 2.50 | 2.94 | 2.04 | 2.80 | 2.31 | 3.04 | 2.50 | 2.05 | 2.59 | 2.07 | 2.66 | 1.96 | 3.09 | | 0.24 |
| 19 | 1.73 | 3.05 | 1.88 | 1.70 | 2.35 | 1.80 | 1.85 | 1.69 | 1.19 | 2.39 | 1.65 | 1.25 | 2.77 | 1.09 | 2.45 | 1.16 | 2.27 | 2.71 | |

discriminatory power, followed by *trn*K (15.8–36%), *mat*K (10.5–16.8%), and *trn*H-*psb*A (5.2–27%). Among the combinations of two barcodes, ITS + *rbc*L had the highest discriminatory success (89.5–100%), whereas that of *mat*K and *rbc*L, which was suggested as the core barcode by CBOL (*CBOL Plant Working Group, 2009*), was only 10.5–25.6%. The three-region combination of ITS + *rbc*L + *trn*H-*psb*A recovered the highest number of monophyletic species (18) in the NJ tree (94.7%). Only five species were successfully discriminated (26.3%) by either the NJ or BI trees using the combination of all four cp markers, i.e., *mat*K + *rbc*L + *trn*H-*psb*A + *trn*K.

## DISCUSSION

### Proposed DNA barcodes for *S.* subg. *Amphilaena*

Among the fragments tested in the present study, ITS obtained a much higher success rate compared with the other loci. In addition, all of the combinations without ITS yielded much lower success rates, regardless of the method used (Table 7). Moreover, the rate of successful PCR (92.7%) was more or less higher for ITS than the other fragments (72.9–91.6%). It has also been reported that this fragment is highly efficient in other Asteraceae genera (*Gao et al., 2010*; *Gong et al., 2016*). However, an intrinsic problem with this fragment is that an individual may have undergone recent hybridization, thereby resulting in multiple mosaic sites (*Li et al., 2011*). In *S.* subg. *Amphilaena*, two species failed to form monophyletic clades in the BI and NJ trees, which could be attributed to the

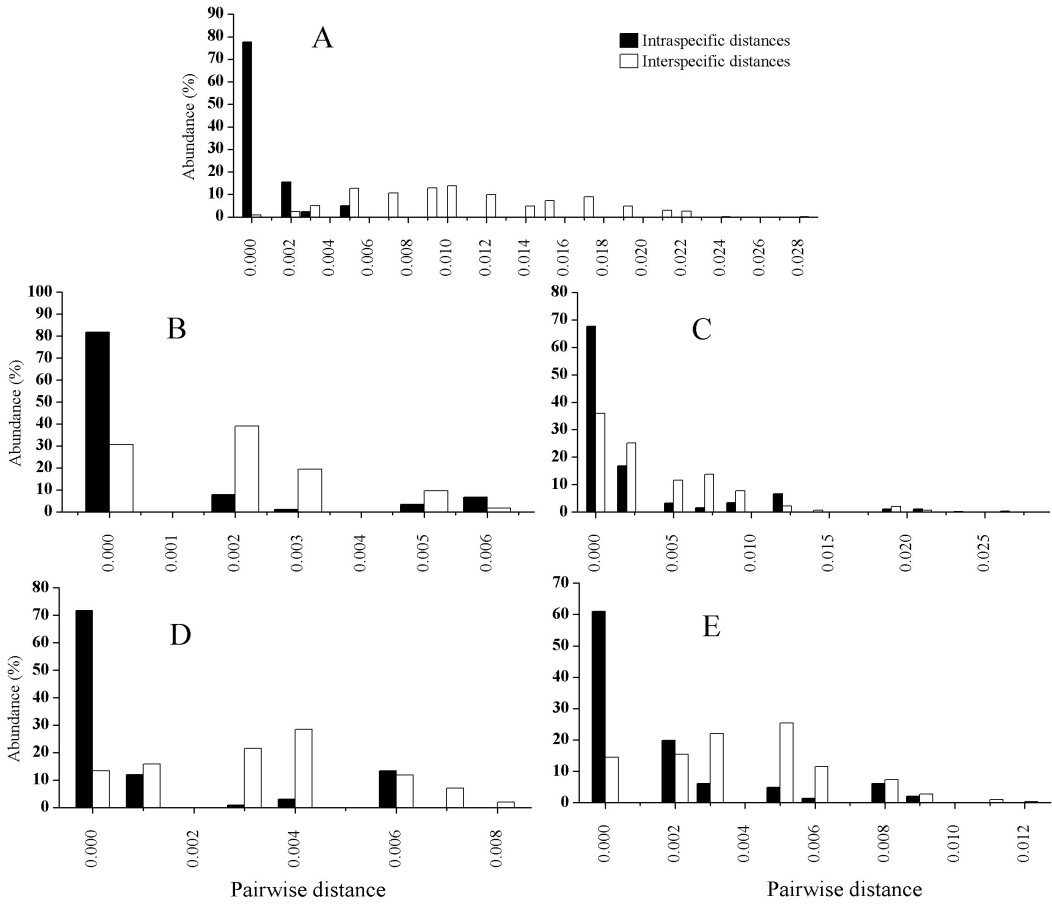

**Figure 2 Relative distributions of intraspecific and interspecific distances calculated with ITS (A), *rbc*L (B), *trn*H-*psb*A (C), *mat*K (D), and *trn*K (E).**

presence of multiple mosaic sites (Fig. 3). However, ITS performed better than the other fragments in *S.* subg. *Amphilaena*, and thus we propose that this fragment should be the first or best choice when selecting only one of the current candidates.

We found that it was difficult to identify the best second choice after ITS. *Trn*K performed much better than *rbc*L in terms of its efficiency when used individually, but its combination with ITS obtained contradictory results, i.e., ITS + *trn*K was inferior to ITS + *rbc*L in terms of efficiency. This contradictory result was unexpected and it is not common in other taxa (*Cao et al., 2010*; *Müller & Borsch, 2005*). We attributed this result to higher degree of congruence of the concatenated sequences of *rbc*L and ITS ($P = 0.12$ for ILD test), in compare to *trn*K and ITS ($P = 0.001$). But it might derive from some other mechanisms, such as the higher rate of mutation for *trn*K that could have caused differentiation within species, but not high enough to form distinct genetic differentiation among species, and thus a failure to cluster as a monophyletic group in line with species (*Naciri, Caetano & Salamin, 2012*; *Petit & Excoffier, 2009*). Therefore, we suggest that using *trn*K alone is problematic and instead we propose to use *rbc*L as complementary to ITS because this
**Table 7  Species resolution using the Best Close Match method and the tree-based method with five barcodes and their combinations.**

| Sequences | Number | Best close match (%) | | | | | BI (%) | NJ (%) |
|---|---|---|---|---|---|---|---|---|
| | | Correct | Ambiguous | Incorrect | No match | Threshold | | |
| ITS | 132 | 93.2 | 6.8 | 0.0 | 0.0 | 0.45 | 84.2 | 84.2 |
| trnK | 125 | 36.0 | 61.6 | 2.4 | 0.0 | 0.91 | 15.8 | 15.8 |
| matK | 125 | 16.8 | 83.2 | 0.0 | 0.0 | 0.56 | 10.5 | 10.5 |
| psbA | 126 | 27.0 | 71.4 | 0.8 | 0.8 | 1.12 | 5.2 | 5.2 |
| rbcL | 125 | 12.0 | 88.0 | 0.0 | 0.0 | 0.63 | 0.0 | 0.0 |
| ITS+trnK | 125 | 98.4 | 0.0 | 1.6 | 0.0 | 0.53 | 79.0 | 84.2 |
| ITS+matk | 125 | 96.0 | 3.2 | 0.8 | 0.0 | 0.36 | 79.0 | 84.2 |
| ITS+psbA | 126 | 96.0 | 4.0 | 0.0 | 0.0 | 0.54 | 84.2 | 89.5 |
| ITS+rbcL | 125 | 100.0 | 0.0 | 0.0 | 0.0 | 0.38 | 89.5 | 89.5 |
| trnK+matK | 125 | 52.0 | 45.6 | 2.4 | 0.0 | 0.72 | 26.3 | 26.3 |
| trnK+psbA | 125 | 52.0 | 44.8 | 3.2 | 0.0 | 0.99 | 21.1 | 21.1 |
| trnK+rbcL | 125 | 37.6 | 60.8 | 1.6 | 0.0 | 0.77 | 15.8 | 15.8 |
| matK+psbA | 125 | 49.6 | 48.8 | 1.6 | 0.0 | 0.77 | 21.1 | 15.8 |
| matK+rbcL | 125 | 25.6 | 74.4 | 0.0 | 0.0 | 0.59 | 10.5 | 10.5 |
| psbA+rbcL | 125 | 30.4 | 68.8 | 0.8 | 0.0 | 0.83 | 10.5 | 5.2 |
| ITS+matK+psbA | 125 | 96.0 | 3.2 | 0.8 | 0.0 | 0.54 | 68.4 | 89.5 |
| ITS+trnK+matK | 125 | 98.4 | 0.0 | 1.6 | 0.0 | 0.54 | 73.7 | 89.5 |
| ITS+trnK+rbcL | 125 | 98.4 | 0.0 | 1.6 | 0.0 | 0.51 | 84.2 | 89.5 |
| ITS+matK+rbcL | 125 | 99.2 | 0.0 | 0.8 | 0.0 | 0.39 | 79.0 | 89.5 |
| ITS+rbcL+psbA | 125 | 100.0 | 0.0 | 0.0 | 0.0 | 0.57 | 79.0 | 94.7 |
| ITS+trnK+psbA | 125 | 98.4 | 0.0 | 1.6 | 0.0 | 0.68 | 79.0 | 89.5 |
| trnK+matK+rbcL | 125 | 52.0 | 45.6 | 2.4 | 0.0 | 0.69 | 26.3 | 26.3 |
| trnK+matK+psbA | 125 | 63.2 | 35.2 | 1.6 | 0.0 | 0.82 | 26.3 | 26.3 |
| matK+psbA+rbcL | 125 | 49.6 | 49.6 | 0.8 | 0.0 | 0.72 | 21.1 | 21.1 |
| rbcL+trnK+psbA | 125 | 55.2 | 41.6 | 3.2 | 0.0 | 0.86 | 15.8 | 21.1 |
| ITS+matK+psbA+rbcL | 125 | 99.2 | 0.0 | 0.8 | 0.0 | 0.57 | 68.4 | 84.2 |
| ITS+matK+psbA+trnK | 125 | 98.4 | 0.0 | 1.6 | 0.0 | 0.64 | 73.7 | 84.2 |
| ITS+matK+rbcL+trnK | 125 | 98.4 | 0.0 | 1.6 | 0.0 | 0.52 | 73.7 | 84.2 |
| ITS+rbcL+trnK+psbA | 125 | 98.4 | 0.0 | 1.6 | 0.0 | 0.66 | 79.0 | 84.2 |
| trnK+matK+psbA+rbcL | 125 | 63.2 | 35.2 | 1.6 | 0.0 | 0.77 | 26.3 | 26.3 |
| ITS+trnK+matK+psbA+rbcL | 125 | 98.4 | 0.0 | 1.6 | 0.0 | 0.64 | 79.0 | 84.2 |

combination could identify all 19 of the sampled species based BCM, and 17 by NJ or BI (89%) (Table 7) (Fig. 4).

The two loci comprising trnH -psbA and matK were affected by the same problem as trnK, with higher mutation rates and barcode efficiencies compared with rbcL when used individually, but lower efficiency when combined with ITS. Thus, their combination with ITS + rbcL failed to significantly increase the success rate and lower results were even obtained in some cases (Table 7). However, among the combinations without ITS, the combination with higher mutation rates was more efficient than those with lower mutation rates, e.g., trnK + trnH-psbA was better than matK + rbcL, which was proposed

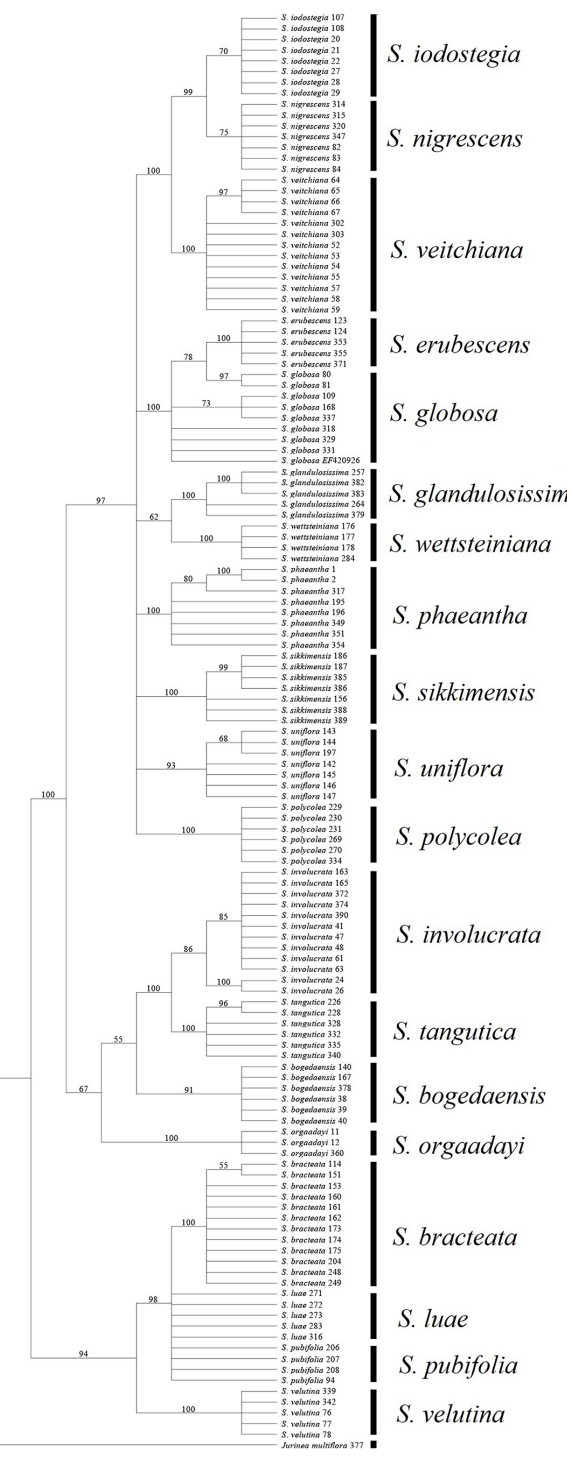

**Figure 3  Phylogenetic tree based on Bayesian analysis of ITS.**

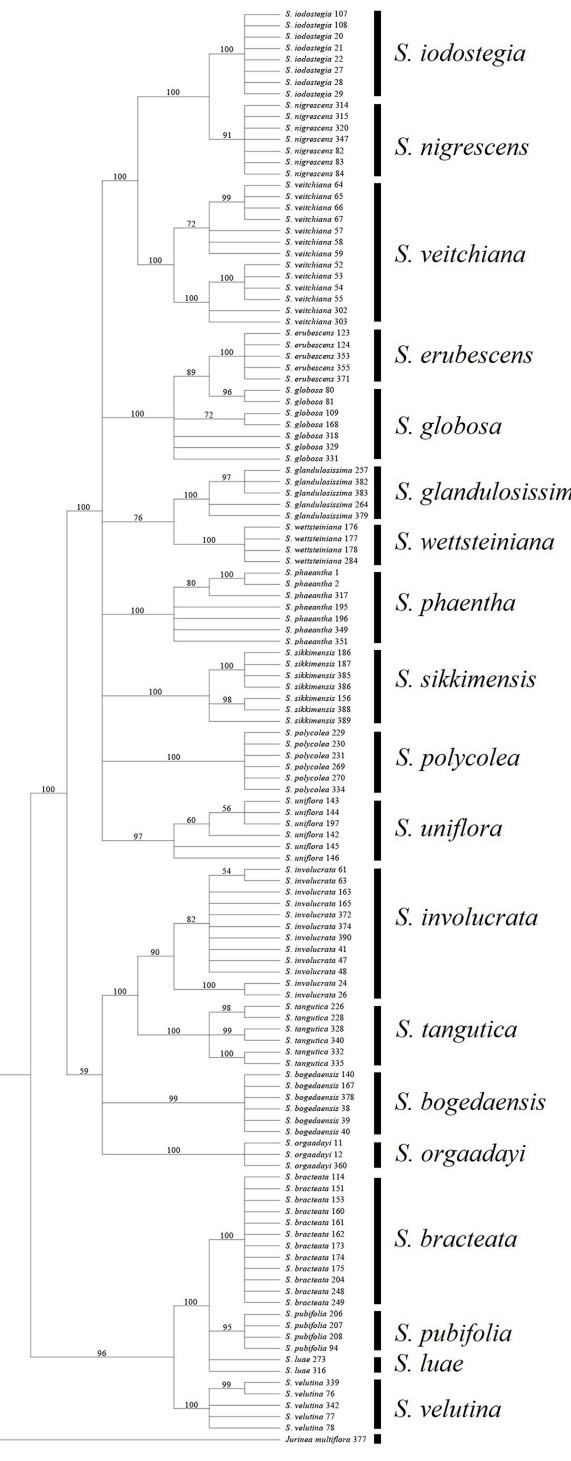

**Figure 4  Phylogenetic tree based on Bayesian analysis of ITS +** *rbc***L.**

previously as the core DNA barcode for plants (*Hollingsworth et al., 2009*). Therefore, if ITS is subjected to hybridization, we propose that the priority order should be the following: *trn*K > *trn*H-*psb*A > *mat*K > *rbc*L. Moreover, the combination with more loci performed better than that with less loci. However, even the combination of all four loci was not sufficient to discriminate each species and new fragments should be considered.

## Insights into taxonomic problems based on DNA barcodes

Most of the analyses failed to identify the species within two groups, i.e., *S. luae* vs. *S. publifolia* and *S. globosa* vs. *S. erubescens* (Figs. 3–5; Table 7). We found that these failures might have been attributable to taxonomic problems. For the first group, we found that *S. luae* was rather heterogeneous in terms of the ITS sequences. Some cp sequences were slightly differentiated compared with *S. velutina*, but the others were closer to those in *S. glandulosissima* or *S. uniflora* (Fig. 5). By contrast, the ITS sequences lacked variance and after excluding the mosaic sites, they were closely related in *S. pubifolia* or *S. bracteata* (Fig. 3). These nuclear-cytoplasmic inconsistencies suggest that hybridization may have occurred among these species.

The second group comprising *S. globosa* and *S. erubescens* was often confused in previous studies because the latter resembles a smaller form of *S. globosa*, which has various forms across its distribution (*Raab-Straube, 2017*). In agreement with the morphology, the genetic distance between the cp sequences within *S. erubescens* was zero whereas that within *S. globosa* was 0.04% (Table 5), which is even larger than that between *S. erubescens* and *S. globosa* (Table 6). The ITS sequences had a very similar pattern and the rich mosaic sites in both species also indicated differentiation accompanying substantial gene flow (*Naciri, Caetano & Salamin, 2012*). Both the BI and NJ methods found that *S. globosa* formed a clade within which *S. erubescens* nested as a monophyletic clade (Fig. 3). Based on these results, we propose that *S. globosa* might be a species with a series of differentiated populations where *S. erubescens* represents one of the most obvious. The current delimitation might need revision on the basis of extensive morphological as well as genetic diversity across the distribution range of both species.

## Identification of the medicinal species and the potential substitutes

All of the known medically important species could be identified using our proposed DNA barcodes, i.e., ITS + *rbc*L or ITS alone (Table 7; Figs. 3–4). Moreover, some species such as *S. bogedaensis*, *S. glandulosissima*, *S. polycolea*, *S. wettsteiniana*, and *S. orgaadayi* could be identified with the cp DNA barcodes (Fig. 5). This high rate of success was unexpected because some species such as the two species in the *S. obvallata* complex (*S. glandulosissima* and *S. sikkimensis*) have been morphologically confused for many years and they were only separated very recently (*Raab-Straube, 2017*). Their distinction is indicative of difference in bioactive components. Therefore, our results caution against their indiscriminating usage in medicine.

Barcode sequences can also help to identify substitutes for medically useful species because closely related species might possibly share the same or similar secondary metabolites and bioactivities (*Zhou et al., 2014*). Thus, we propose that nine of the 15

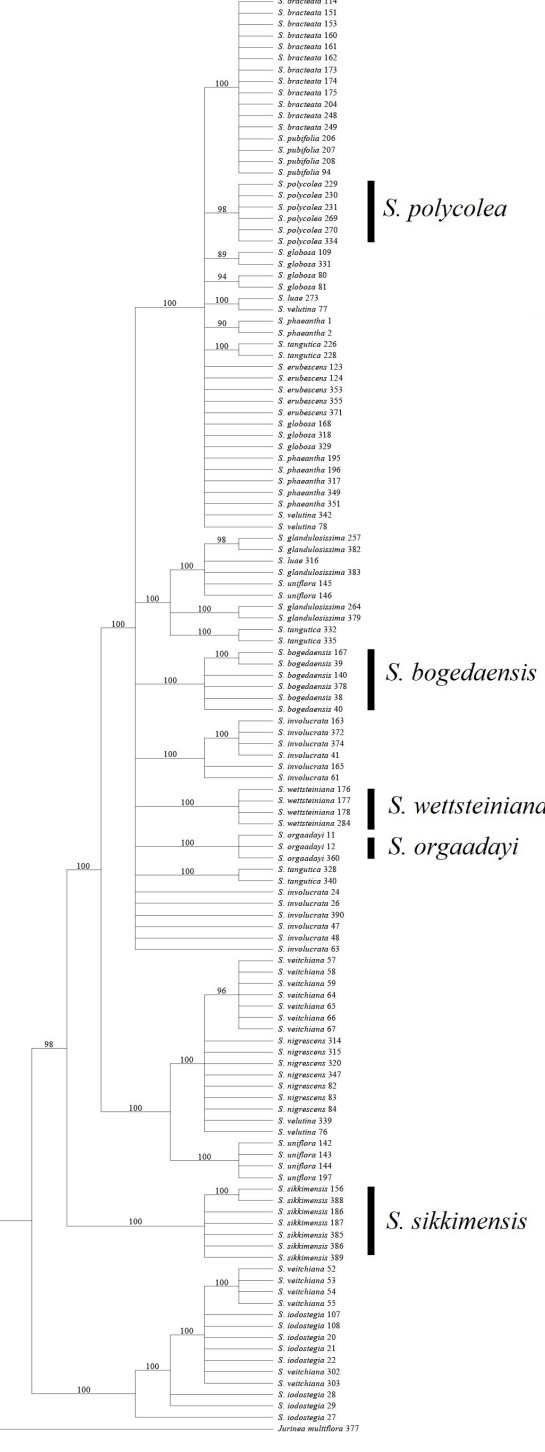

**Figure 5 Phylogenetic tree based on Bayesian analysis of *trn*K + *mat*K + *psb*A + *rbc*L.**

medically useful species might be substituted by their close relatives according to the molecular phylogenetic context. Six of these species, which formed three groups, are also morphologically similar, i.e., *S. involucrata* and *S. orgaadayi* or *S. bogedaensis*, *S. globosa* and *S. erubescens*, and *S. wettsteiniana* and *S. glandulosissima* (Fig. 3) (*Raab-Straube, 2017*). Among the remaining three species, *S. bracteata* appears to be closely related to *S. pubifolia* whereas *S. iodostegia* and *S. nigrescens* are closely related to each other according to phylogenetic tree (Fig. 3). These affinities were not expected according to their morphology, but they are possibly due to convergent evolution or radiation in *Saussurea* (*Wang et al., 2009*). Secondary metabolomes or bioactivities are wanted to confirm their similarity.

## CONCLUSION

Based on the sequence statistics, inter- and intraspecific distances, SPIDER, and phylogenetic analyses, it is concluded that internal transcribed spacer (ITS) + *rbc*L or ITS + *rbc*L + *psb*A-*trn*H could distinguish all of the species, while the ITS alone could identify all of the 15 medical plants. However, the species identification rates based on plastid barcodes were low, i.e., 0% to 36% when analyzed individually, and 63% when all four loci were combined. Thus, we recommend using ITS + *rbc*L as the DNA barcode for *S.* subg. *Amphilaena* or the ITS alone for medical plants.

## ACKNOWLEDGEMENTS

We are grateful to Jian-Quan Liu, Zhong-Hu Li, Yi-Xuan Kou, Fu-Shen Yang and Hiroshi Ikeda for helping with our field investigation.

### Funding
This study was supported by the National Natural Science Foundation of China (81274024). The funders had no role in study design, data collection and analysis, decision to publish, or preparation of the manuscript.

### Grant Disclosures
The following grant information was disclosed by the authors:
National Natural Science Foundation of China: 81274024.

### Competing Interests
The authors declare there are no competing interests.

### Author Contributions
- Jie Chen conceived and designed the experiments, performed the experiments, analyzed the data, contributed reagents/materials/analysis tools, prepared figures and/or tables, authored or reviewed drafts of the paper, approved the final draft.
- Yong-Bao Zhao performed the experiments, contributed reagents/materials/analysis tools, prepared figures and/or tables.

- Yu-Jin Wang contributed reagents/materials/analysis tools, authored or reviewed drafts of the paper, approved the final draft.
- Xiao-Gang Li approved the final draft.

## Data Availability

All of the sequences used in this article are registered in GenBank: accession numbers MH003704 to MH003835 for ITS and MH070616 to MH071120 for chloroplast regions.

## Supplemental Information

Supplemental information for this article can be found online at http://dx.doi.org/10.7717/peerj.6357#supplemental-information.

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
