# Peer review of "Identification of species and materia medica within Saussurea subg. Amphilaena based on DNA barcodes"

_PeerJ, doi:10.7717/peerj.6357_

## Round 0.1 · original submission · Minor Revisions

Dear Authors,

We have requested opinions of three experts. As you can see from the attached reviewers' comments, there are some concerns about the clarity of the manuscript and several minor points need to be modified or clarified. Please revise the manuscript as suggested by reviewers or provide a rebuttal. We look forward to reading your revised manuscript.

Sincerely,
Marta Kostrouchova

Reviewer 1 ·

Basic reporting

The authors should revise the English grammar of the manuscript.

Experimental design

Appropriate.

Validity of the findings

Appropriate and accurate.

Additional comments

Saussurea is an important genus in Asteraceae with high species diversification in high mountains in northern hemisphere. The authors made a detailed and thorough barcoding analyses on an interesting group of subg. Amphilaena, which is traditional and very popular as medicine uses. All the commonly used markers have been sequenced for barcoding analyses and species identification with comprehensive sampling including both populations and individuals. Data analyses and explanations are enough and excellent. This study is fell well within the scope of the journal, it is a good paper to be accepted.
A few minor issues are given below:
1. Line 116: a having more than two populations ?
2. Line 119: to my knowledge, matK is included in trnK and only matK is commonly used for barcoding. Since only a small fragment is only sequenced in barcoding analyses, trnK here is only a different short fragment from matK. Please clarify the use of trnK and its difference from matK.
3. Line 131: our sampled ?
4. Line 197: an the individual ?
5. Line 484 Table 2: what’s meaning for numbers after voucher numbers in the bracket.

·

Basic reporting

The authors collected extensive samples from Saussurea in the field and they sequenced five DNA barcode candidates in chloroplasts (matK, trnH-psbA, trnK, and rbcL) and the nuclear ITS. This manuscript is well written, I suggested that this work should be accepted after minor revision.

Experimental design

I suggest authors should utilize incongruence length difference and partition homogeneity test when you use more than one barcode markers to discriminate species.

Validity of the findings

If possible, authors should collect some commercial herb in market to test their identity according to Pharmacopoeia of People's Republic of China.

Reviewer 3 ·

Basic reporting

Chen et al. have done a good job to test five candidate barcode in delimitation of medical plants in Saussurea subg. Amphilaena, the results are clear and the conclusions are umambiguous, and their English are mostly satisfied. They are familiar with taxonomy and references related to Saussurea, and their organization of the article are professional.

Experimental design

The experimental design of this work is sufficient to address key questions the authors want to ask.

Validity of the findings

The data that the authors presented in this manuscript is robust. Usually barcoding researches sample 3 individuals per species, and in this study they sampled ca. 8 individuals per species, which is more than enough. The author well addressed their key questions by running a series of analyses and the key conclusions they made are objective, i.e. well supported by their data and results.

Additional comments

Chen et al. have done a good job to test five candidate barcode in delimitation of medical plants in Saussurea subg. Amphilaena, the results are clear and the conclusions are umambiguous, and their English are mostly satisfied. They are familiar with taxonomy and references related to Saussurea, and their organization of the article are professional. The data that the authors presented in this manuscript is robust. Usually barcoding researches sample 3 individuals per species, and in this study they sampled ca. 8 individuals per species, which is more than enough. The author well addressed their key questions by running a series of analyses and the key conclusions they made are objective, i.e. well supported by their data and results. However, I have some comments which may be helpful to improve the manuscript.
(1) At the end of the Introduction as well as in the paragraph before Conclusion, the authors discussed about possible substitute of closely related medical plants. I think on this point, the author should be cautious. It would be a good strategy to put forward potential substitute species, yet it is highly necessary to warn the readers that extraction and validation of medical components should be done before such substitution be made. The other way is that, your DNA barcoding method could be use to refrain substitution of well-known medical species with species that is common yet less effective medically.
(2) In L197-191, the author claimed that lower gene flow of rbcL compare to ITS and higher mutation rate of trnK compare to rbcL is the reason why combination of ITS+rbcL perform better than ITS+trnK. However, my opinion is that, maybe weaker incongruence between ITS and rbcL than between ITS and trnK is the most important issue, i.e. ITS signal is much stronger than rbcL, and hence rbcL work as a complementary part, but both ITS and trnK signals are strong and thus lead to a conflict between barcodes. This can be tested by conducting Partition homogeneity testing that is performed in PAUP*. Just do the test for ITS vs rbcL and ITS vs trnK would be sufficient.

Minor concerns:
L69: change 'with' to 'show'?
L70: distantly related 'according to molecular phylogeny'?
L86: is is necessary to put 'S. subg. Amphilaena' after 'useful'?
L90: second-class 'protected plants in China'?
L91: 'of' may be replaced with a coma
L93: What do you mean? Do you mean identifying substitutes for medical usages is urgently needed? Or do you mean identifing fake medical materials (cheap substitute expensive) is urgently needed?
L111: I think nearly all plants you involved have been reported to have medical value, how can you find new plants? I suppose that you should have to involve previously not-recorded-as-medical plants to mine new medical plant species.
L148: I think you should tell PCR success rate and sequencing success rate separately. The current presentation is wrong and misleading. The PCR and sequencing success rate should be the product that multiply the PCR success rate and the sequencing success rate.
L175: I do not agree with this. The PCR successful rate is significantly higher than trnK and trnH-psbA, yet more or less close to rbcL and matK.
L187-188: please clarify;
L190: do you mean 'higher intraspecific differentiation between populations'?
L204-206: please clarify;
L227-228: Please rephrase to improve clarity;
L231: is 'potential' better than 'possible'?
L238-240: I think you need more evidence here. Or else, warn the authors that you are infer and assume. Plus, this is slightly contradict with the next paragraph where you said closely related species share the same or similar metabolimes and bioactivities.
L251-254: I do not think this is a good ending of the paragraph. A better ending would be suppose that these complex you proposed could be further checked with their secondary metabolimes and bioactivities, so as to make sure some plants can substitute the other, if cultivation is not possible.
In addition, I would like to recommend the author to add a figure to show some beautiful photos of some beautiful species of this section in Saussaurea. I am sure that will attract attention from many non-taxonomist readers!

---

## Round 0.2 · accepted · Accept

I have requested the opinions of three expert reviewers and all of them seem to find that the manuscript was carefully revised and is now accepted for publication.

With kind regards,
Marta Kostrouchova
Academic Editor , PeerJ

# Reviewer 1 ·

Basic reporting

Clearly.

Experimental design

Reasonable.

Validity of the findings

Accurate.

Additional comments

I have checked the manuscript again, and I found that the authors have carefully revised the manuscript according to my previously suggestions and comments. So I recommend it acceptance in its current form.

·

Basic reporting

I think the authors have added these analysis i mentioned.

This manuscript should be accepted.

Experimental design

no

Validity of the findings

no

Additional comments

no

Reviewer 3 ·

Basic reporting

N/A

Experimental design

N/A

Validity of the findings

N/A

Additional comments

I am happy that the authors addressed all concerns that I raised in last round of review carefully and in a point-by-point manner. I think the manuscript reads stronger than last version, and I believe this work will attract interests from many readers working on medical plants, Asteraceae, DNA barcoding and high altitude flora, among others.